# Abdominal Wall Block Decreases Intraoperative Opioid Con-Sumption by Surgical Pleth Index-Guided Remifentanil Administration in Single-Port Laparoscopic Herniorrhaphy: A Prospective Randomized Controlled Trial

**DOI:** 10.3390/ijerph192316012

**Published:** 2022-11-30

**Authors:** Eung Don Kim, Youngin Lee, Segyu Choi, Hyein Lee, Chaeryeon Ohn, Woojin Kwon

**Affiliations:** Department of Anesthesiology and Pain Medicine, Daejeon St. Mary’s Hospital, College of Medicine, The Catholic University of Korea, Daejeon 34943, Republic of Korea

**Keywords:** acute pain services, interfascial plane block, laparoscopic hernia repair, surgical pleth index, rectus sheath block, quadratus lumborum block, intraoperative monitoring, pain measurement

## Abstract

Abdominal wall blocks (AWBs) can reduce pain during surgery and lessen opioid demand. Since it is difficult to know the exact level of intraoperative pain, it is not known how much the opioid dose should be reduced. In this study, using the surgical pleth index (SPI), which indicates pain index from sympathetic fibers, the amount of remifentanil consumption was investigated. We conducted single-port laparoscopic hernia repair in 64 patients, as follows: the regional block group (R group) was treated with AWB, while the control group (C group) was only subjected to general anesthesia. In both groups, the remifentanil concentration was adjusted to maintain the SPI score between 30 and 40 during surgery. The primary parameter was the amount of remifentanil. A total of 52 patients completed the study (24 in the R group, 28 in the C group). The remifentanil dose during surgery was decreased in the R group (29 ± 21 vs. 56 ± 36 ng/kg/min; *p* = 0.002). Visual analogue scale score and additional administrated analgesics were also low in the R group. As such, AWB can reduce the remifentanil dose while maintaining the same pain level.

## 1. Introduction

Recent truncal interfascial plane block techniques, such as transversus anterior plane (TAP) block, quadratus lumborum block (QLB), rectus sheath block (RSB), and erector spinae plane (ESP) block, have attracted attention as a variety of ways to reduce postoperative pain [1,2,3,4]. However, these nerve block techniques are not able to completely mitigate all pain, so they are established as only some of the pain control methods available [2]. There are many controversies as to whether these nerve block methods can manage not only somatic pain but also visceral pain [5,6]. When a TAP block or QLB is performed, a pain reduction effect is seen even if a successful dermatome block is not confirmed by the pinprick test and cold sensory test [7]. Since visceral pain is vague and difficult to localize, measuring the amount of pain on the visual analogue scale (VAS) or opioid consumption may yield inaccurate results. Furthermore, although the range of local anesthetics spread with dyes in cadaver studies was visually confirmed, some studies suggest there is a discrepancy regarding where the sensory loss occurred [8,9,10,11]. A tiny amount of local anesthetic spreading to some unmyelinated nerves cannot be confirmed, and it is impossible to know whether the local anesthetic has shifted into living tissue. Because the expansion of local anesthetic is decided by the tension of the fascial layers and muscle, it cannot be replicated in cadaver studies [12,13]. An abdominal wall nerve block may attenuate pain during single-port laparoscopic totally extraperitoneal (SP TEP) inguinal hernia repair, thus, reducing the degree of opioid consumption [14,15]. However, to the best of our knowledge, there has been no objective research conducted on the extent of decrease in opioid reduction and how much is appropriate.

Since the patient’s pain cannot be directly expressed during general anesthesia, the pain level is indirectly discerned from parameters, such as heart rate, blood pressure, or electroencephalography [16]. However, these indicators were not found to be closely related to pain. Recently, several methods (SPI, analgesia nociception index or ANI, pupillometry, skin conduction, and CARDEAN) have been introduced to determine the degree of pain during surgery [17]. Among them, the SPI adopts saturation pleth graphs to determine the balance between nociception and antinociception, and it indicates the score between 0 and 100 points. For clinical use, an SPI score of close to 100 indicates a very high-stress level, whereas one close to 0 points indicates a very low stress level. 

The surgical pleth Index (SPI) wa” dev’loped in 2007 as a method for measuring pain via plethysmography analysis through peripheral (sympathetically mediated) vasoconstriction and cardiac autonomic tone [18]. So far, it is believed to be the most reliable method for analyzing pain levels during surgery [19].

In this research, among patients with unilateral hernia surgery, preoperative abdominal wall nerve block was performed before surgery, and a similar SPI grade between a nerve block group and a control group was maintained by increasing or decreasing remifentanil concentration according to pain.

We sought to quantify the difference in the degree of pain reduction due to nerve block triggered by remifentanil consumption between the two groups.

We hypothesize that intraoperative opioid dosage would be reduced while maintaining the same nociception level by measuring SPI through preoperative abdominal wall block in SP TEP patients.

## 2. Materials and Methods

This paper was approved by the institutional review board of Daejeon St. Mary’s Hospital in Daejeon, South Korea (protocol no. DC13OISI0076, approved 17 May 2018). The project was registered with the Clinical Research Information Service (no. KCT0001461).

The subjects were those who underwent unilateral SP TEP herniorrhaphy. On the day of surgery, patients were hospitalized and their consent for inclusion was obtained after an explanation of the study was provided.

Eligible subjects were those 18 to 70 years old with an American Society of Anesthesiologists status classification of grade 1 or 2. Patients with cardiovascular, respiratory, neurologic, or kidney disease, prematurity, developmental delay, fentanyl or remifentanil allergy, use of other analgesics, anticonvulsants, or psychiatric drugs, or a severely anxious or uncooperative disposition, were excluded.

The control group (C group) was subjected to conventional general anesthesia per the usual protocol using an inhaled anesthetic. The regional block group (R group) underwent both rectus sheath block and quadratus lumborum block following the induction of anesthesia, and then herniorrhaphy surgery was initiated. 

### 2.1. Randomization

As indicated above, the study subjects were stratified into the R and C groups; a computer program (R; R Foundation for Statistical Computing, Vienna, Austria) created a random number table and number assignments were sealed in individual envelopes. 

For blinding, three anesthesiologists adopted independent roles, as follows: one initiated anesthesia and conducted the nerve blocks (the ‘practitioner’), one maintained anesthesia (the ‘maintainer’), and the third evaluated pain and various data after surgery (the ‘evaluator’). 

During the surgical procedure, when the patient entered the operating theatre, the practitioner (first anesthesiologists) opened the sealed randomized bag and identified whether were assigned to the C group or the R group. If the patient was assigned to the R group, general anesthesia was performed, followed by RSB and QLB, while, if the patient was assigned to the C group, only general anesthesia was performed. 

At this point, the practitioner was replaced with the ‘maintainer,’ the second anesthesiologist who ensured that anesthesia was established stably prior to the start of surgery. Unlike the first anesthesiologist, this person was blinded to the assigned group. During surgery, this anesthesiologist recorded and controlled the remifentanil dose (target-controlled infusion mode) under SPI guidance and collected extubation-related data. To obtain an accurate pain score in the recovery room, this anesthesiologist, still blinded to the group, transferred the patient to the recovery room and examined the pain, nausea, and vomiting scores. 

Finally, at this stage, the third anesthesiologist became involved, and blinding to the group assignment was dismissed after examining the pain score. If the patient was in the R group, the range of the nerve block and other recovery-related data was investigated.

Drugs, such as midazolam, fentanyl, and glycopyrrolate that may affect entropy or the SPI score, were avoided as premedications. 

The SPI scores and patient blood pressure and entropy were analyzed by downloading to the computer every 10 s using the S5 collect (GE healthcare, US) program.

After entering the operating room, the patient’s blood pressure, electrocardiogram, heart rate, oxygen saturation, entropy, and SPI were measured by a patient monitor (CARESCAPE™ B850; GE Healthcare, Chicago, IL, USA). 

Both groups underwent the same method of starting anesthesia. General anesthesia was initiated using propofol 2 mg/kg and rocuronium bromide 0.6 mg/kg. In the ventilation setting, the tidal volume was 8 mL/kg body weight, and the respiratory rate was adjusted such that the ETCO2 was maintained at 30 to 35 mmHg, while PEEP was not applied (Aisys™ CS2; GE Healthcare, Chicago, IL, USA). After intubation, general anesthesia was achieved by inhalation of the inhalation anesthetic sevoflurane (1.5–2.5%). The concentration of sevoflurane was adjusted to between 0.8 and 1.2 minimal alveolar concentration (MAC) to maintain an appropriate anesthetic depth level of between 40 to 60 points using entropy monitoring (Entropy™ module; GE Healthcare, Chicago, IL, USA). The reason for using sevoflurane was that it is more sensitive to the SPI score [20]. Remifentanil (Ultiva; GlaxoSmithKline, Brentford, UK) was used as an additional anesthetic to reduce pain. 

In both groups, the amount of sustained infusion of remifentanil was adjusted according to the SPI score. 

After all skin sutures were completed, the second anesthesiologist stopped delivery of the inhalation anesthetic and remifentanil infusion; subsequently, 30 mg of ketorolac was administered as an analgesic and 0.3 mg of ramosetron was given to prevent N/V. 

### 2.2. Spi-Guided Remifentanil Infusion

Remifentanil was injected into a syringe pump (Injectomat TIVA Agilia®; Fresenius Kabi AG, Bad Homburg, Germany) that was programmed in target-controlled infusion (TCI) mode. The target concentration is chosen based on the effect-site concentration (Ce).

When beginning the anesthesia, we started with 3 ng/kg body weight/min. After tracheal intubation, however, this was lowered to 1.5 ng until the surgery was started (Figure 1).

The remifentanil was adjusted to achieve a surgical pain score of between 30 and 40 points. Although there are several papers suggesting cutoff values, we chose 30 points based on the research of Ledowski [21]. When the surgical pain score rose higher than 40 points for 20 s, the Ce was increased by 0.5 [22,23]. In this case, we re-evaluated after five minutes, attempting to keep the value down below 40 points to determine whether we could reduce the Ce by 0.5. If necessary, it could be increased again up to 7.0 ng/kg/min. However, if the score did not drop below 40 points even when the dosage was increased to 7.0 ng/kg/min, the case was excluded from this experiment.

Conversely, when the surgical pain score fell below 30 points for 20 s, the effective concentration of the pump was decreased by 0.5. We re-evaluated again after five minutes and attempted to maintain the value once it rose to 30 points or more or decreased by 0.5 again if still under 30 points. It can be reduced all the way to a minimum of 0.5 ng/kg body weight/min. 

Hypotension was defined as when the blood pressure was reduced by more than 30% relative to the average blood pressure at the time of admission. An injection of phenylephrine 50 mcg was conducted because it is less affected by SPI in comparison with other inotropes [20]. 

If hypotension persists even after two doses of phenylephrine, 10 mg of ephedrine was administered, and the patient was excluded from the study. Likewise, when a blood pressure that was 30% higher than that recorded when entering the room (hypertension) was noted, 0.5 mg of nicardipine was given. If the blood pressure remained high after nicardipine was given twice, the case was excluded from the study [20].

### 2.3. Abdominal Wall Block Method in The R Group

Two abdominal wall blocks were used to reduce pain during surgery. All nerve blocks were performed by an expert with 10 years of experience in regional anesthesia. From among the various types of QLB, lateral QLB (type I) was performed (Figure 2A). A lateral QLB approach can block the ilioinguinal and iliohypogastric nerves from T12 and L1 to control pain from the inguinal area [2,24]. Here, 15 mL of 0.3% ropivacaine was injected. The RSB is a nerve block method that causes umbilical area sensory loss and is an effective technique in SP TEP herniorrhaphy surgery or umbilical hernia surgery. In total, 15 mL of 0.3% ropivacaine was injected on each side (Figure 2B).

In the recovery room, the evaluator attempts a pinprick test on the umbilical area to check the block success. If there was no sensory loss within 5 cm above and below the umbilicus, it was considered a block failure and was excluded from the study.

### 2.4. Surgical Procedure and Ward Management 

The whole procedure was not different from that of SP TEP inguinal hernia repair and involved 3 experienced surgeons who have performed more than 300 operations. During surgery, head up and head down could also affect SPI, so both were limited within around 15 degrees [25]. 

In the recovery room, when the patient’s recovery score reached 10 on the modified Aldrete score and 9 or higher on the postanesthesia discharge scoring system (PADS), the patient was transferred to the ward and the recovery time was recorded.

After surgery, patients followed the routine acute pain service protocol for SP TEP. For those patients with a VAS score of six points or higher in the ward, 100 mg of tramadol was given. If the vital signs were fine at six hours after surgery, the patient was discharged on the same day.

### 2.5. Statistical Analysis and Endpoints

The primary endpoint of this study was the amount of remifentanil per body weight per minute given during surgery.

Secondary parameters included remifentanil adjustment values, extubation time, and resting and coughing VAS scores in the recovery room, four hours after surgery, and at discharge from the ward. The number of additional painkillers administered was also recorded. Nausea and vomiting were recorded according to a 4-point scale routinely used in our postanesthesia care unit (0 = no nausea, 1 = occasional nausea, 2 = persistent nausea requiring treatment, and 3 = vomiting) [26]. 

The number of subjects was calculated based on 10 patients in each group in the pilot study. In the pilot study, the remifentanil consumption per minute of the C group was 0.065 ± 0.024 mcg/kg body weight/min, the remifentanil consumption per minute of the R group was 0.0443 ± 0.023 mcg/kg body weight/min, and the effect size of the two groups was 0.884. Therefore, the allocation ratio of the two groups was set as 1:1, with a Student’s *t*-test, two-sided test, *p* = 0.05, and a power value of 0.8. Twenty-two patients were required per group, and 32 patients per group were chosen, considering a 30% dropout rate. Ultimately, a total of 64 patients participated in the experiment.

Using the Statistical Package for the Social Sciences statistics program (IBM Corp., Armonk, NY, USA), a t-test was used to compare average values of continuous variables, such as remifentanil consumption, VAS score, and recovery time between the two groups. The chi-squared test or Fisher’s exact test was adopted to conduct comparisons, represented as ratios.

## 3. Results

Data collection started in May 2018 and ended in April 2020 at Daejeon St. Mary’s Hospital.

Here, 9 out of a total of 64 patients refused to enter the study, so 55 patients participated in the study. Later, three of the R group patients were excluded for the following reasons: one subject experienced arrhythmia during the operation, and as a result, the SPI score was unreliable; one showed an increased amount of CO_2_ while emphysema was occurring, and their heart rate increased more than 100 times; and one subject did not achieve an SPI score of below 40 points even though the remifentanil was raised above 7.0 ng/kg body weight/min. Thus, the R group included 24 patients and the C group included 28 patients (Figure 3).

There were no significant differences between the two groups in terms of height, weight, American Society of Anesthesiologists status classification, body mass index, initial V/S, initial SPI score, and the percentage of professors between the two groups (Table 1). As a result of testing the effect of the block in the R group in the recovery room, it was confirmed that appropriate nerve block was achieved in 100% of patients.

The primary parameter—the amount of remifentanil per kg body weight per minute given during surgery—decreased to 29.5 ± 21.2 ng/kg body weight/min in the R group and 55.7 ± 35.5 ng/kg body weight/min in the C group (*p* = 0.002) (Table 2) (Figure 4). In addition, the number of remifentanil adjustments during surgery was also decreased in the R group (3.0 ± 1.7 vs. 4.6 ± 2.2; *p* = 0.007). The mean SPI score during surgery was not different between the two groups (27.6 ± 7.6 vs. 28.5 ± 9.6; *p* = 0.76). There was no difference in extubation time, mean blood pressure, or pulse rate between the two groups. 

The VAS scores were decreased during both resting and cough before discharge in the R group. The administration of additional analgesics was significantly reduced until four hours after surgery, but there was no difference between the groups at discharge. Overall, there were no differences in other parameters between the two groups (Table 3).

## 4. Discussion

This study examines how much opioid consumption decreases when maintaining a similar SPI score between two groups with and without nerve block. Remifentanil consumption decreased by 47% in the regional block group (29.5 vs. 55.7 mcg/kg body weight/min) during SP TEP hernia repair.

A method to obtain objective pain level data to support the provision of adequate analgesics during surgery was developed in the early 2000s but has not yet entered the standard clinical practice repertoire of anesthesia as an essential patient monitoring approach.

Intraoperative as well as postoperative pain control is one of the most important parts of anesthetic care. If pain is not properly managed, the sympathetic nervous system is activated, and the cardiac workload and SVR increase [26,27]. 

The SPI algorithm is not known definitively because of GE Healthcare’s proprietary claim, but it is roughly known to be as follows: SPI = 100 − (0.33 × HBInorm + 0.67 × PPGAnorm)(1)
where HBInorm is the normalized heartbeat interval and PPGAnorm is the normalized photoplethysmographic pulse wave amplitude.

Since SPI is influenced by the autonomic nervous system, several factors affecting the autonomic nervous system prevent accurate values from being obtained. Inotropes, hypotensive agents [28,29], head-down position [25], hyperinflations in laparoscopic surgery [30], the difference of inhalation anesthesia [20], anesthetic technique [31], blood volume status [32], and patient’s age [33] may affect the SPI score.

Not surprisingly, if too deep a level of anesthesia or too high a concentration of analgesic is used, the SPI response may be attenuated [16]. A study of patients with moderate to severe pain reported that SPI was more accurate than heart rate and mean artery pressure, but the predictive accuracy overall remained low [34]. 

On the contrary, many papers have positively evaluated the effectiveness of SPI. In an early study of SPI, Struys said that, for noxious stimulus, it is more accurate than the electroencephalogram analysis state and response entropy of electroencephalogram analysis [35]. In a study on vascular surgery, where ANI was examined, remifentanil consumption during surgery, postoperative pain, and rescue analgesia were decreased [23]. Elsewhere, ANI, SPI, and pupillometry responded more sensitively to pain than heart rate and MAP, suggesting that the bispectral index (BIS) was not recognizable as an indicator of analgesia [16]. 

This study assessed how much remifentanil usage was reduced by using SPI when pain decreased due to an abdominal wall nerve block. Through this study, it was found that the effect of a nerve block before surgery could be quantitatively measured, thus, reducing the remifentanil infusion rate by half and providing an equivalent analgesic effect. Unlike other laparoscopic surgeries, SP TEP surgery establishes pneumoperitoneum between the abdominal rectus muscle and the peritoneum rather than in the abdominal cavity, so pain characterization is expected to be somatic pain more so than visceral pain in the early stages of surgery. Therefore, the authors performed an RSB that can block the umbilical area where the laparoscopic port enters [36] and a lateral QLB (type I) able to block the ilioinguinal (L1), iliohypogastric (L1), and subcostal (T12) nerves that distribute the inguinal area [2,24]. There is also a case report in which surgery was performed with only PNB (peripheral nerve block) without general anesthesia [37]. However, more than moderate sedation was necessary in that context; when dissection of the surrounding tissues to apply mesh to the area where the hernia occurred was completed, the patient’s agitation and urinary symptoms increased. In our study, if all of the pain was resolved with truncal blocks, there would be fewer adjustments of the remifentanil concentration during surgery. In the R group, the average adjustment was three times, and it is thought that the occurrence of pain during surgery changed the SPI score.

As a result of discussions between the authors of this study, SPI fluctuations appeared mainly during the latter part of the surgery, and this is considered to be consistent with the aforementioned case report. Therefore, even though QLB and RSB blocks reduced the use of drugs in SP TEP surgery, the authors believe that some somatic or visceral pain that cannot be covered by truncal interfascial plane block occurred. 

The postoperative nausea vomiting (PONV) grade was found to have no significant difference between the two groups. Furthermore, there were very few people who complained of nausea in both groups. The PONV risk factors are female, history of PONV, non-smoker, younger age, volatile anesthetics, nitrous oxide, postoperative opioids, and laparoscopic surgery [38]. Subjects had low probability of nausea vomiting, because the patients were mainly elderly and male, were not undergoing long operation times (under an hour), and the surgery procedure did not make a pneumoperitoneum directly in the abdominal cavity.

Interestingly, when the local anesthetic was injected into the medial side, it may interfere with the surgical field of view, so the injection was performed in the lateral side as much as possible. 

There are some limitations to the present research. Inguinal hernia disease affects a high proportion of men, and cases of this condition in females were lacking in this study. More data are needed to mitigate this gender imbalance in the future.

Furthermore, it was not clear whether the effect of pain reduction was due to PNB or the local anesthetic effect absorbed systemically, because no sham block was performed. 

In this paper, when remifentanil infusion of 7.0 ng/kg body weight/min or more was eliminated from the study, on the contrary, it did not drop even if it fell below 0.5.

## 5. Conclusions

To conclude, rectus sheath block and quadratus lumborum block can reduce remifentanil use by half during a single laparoscopic hernia operation. 

Further investigations are needed to understand the relationship between the degree of neural blockade through a regional block and SPI monitoring.

## Figures and Tables

**Figure 1 ijerph-19-16012-f001:**
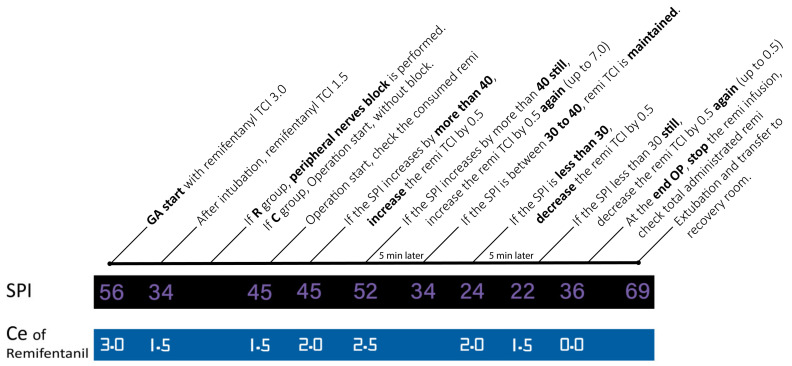
SPI-guided remifentanil infusion. Abbreviations are as follows: SPI, surgical pleth index; OP, operation; remi, remifentanil; TCI, target-controlled infusion; GA, general anesthesia; R, regional group; C, control group; Ce, concentration of effector site.

**Figure 2 ijerph-19-16012-f002:**
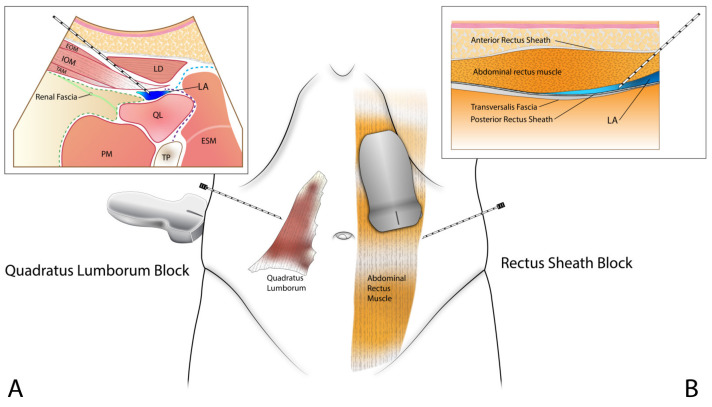
Abdominal wall blocks conducted in the R group. Schematic diagram. (**A**) Ultrasound guided quadratus lumborum block (type I). The target of the injection point is the fascial plane between the quadratus lumborum muscle and the transversus abdominis muscle. (**B**) Ultrasound guided rectus sheath block. The target of the injection point is the fascial plane between the posterior rectus sheath and the abdominal rectus muscle. Abbreviations are as follows: LA, local anesthetic; EOM, external oblique muscle; IOM, internal oblique muscle; LD, latissimus dorsi muscle; PM, psoas muscle; QL, quadratus lumborum muscle; TP, transverse process.

**Figure 3 ijerph-19-16012-f003:**
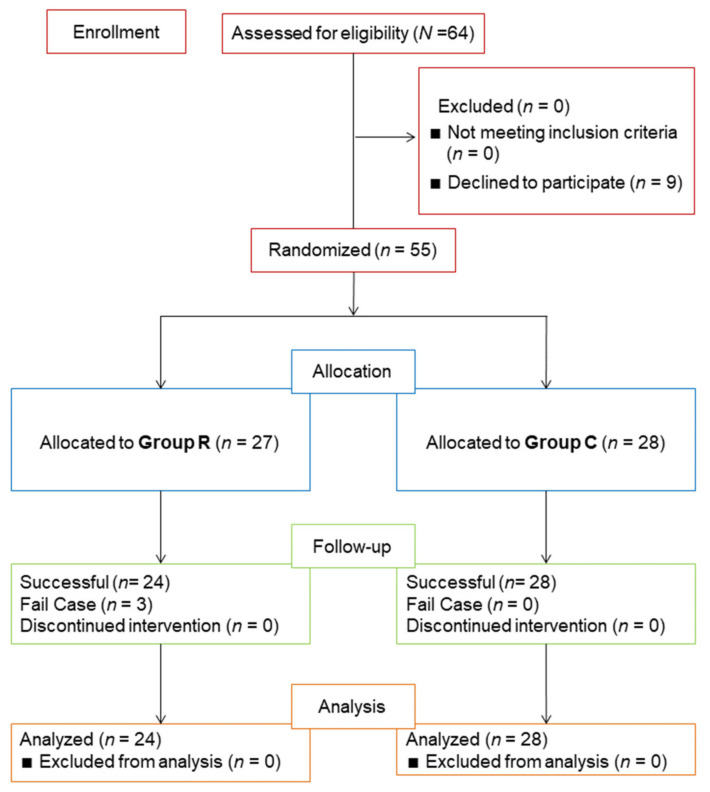
Group allocation.

**Figure 4 ijerph-19-16012-f004:**
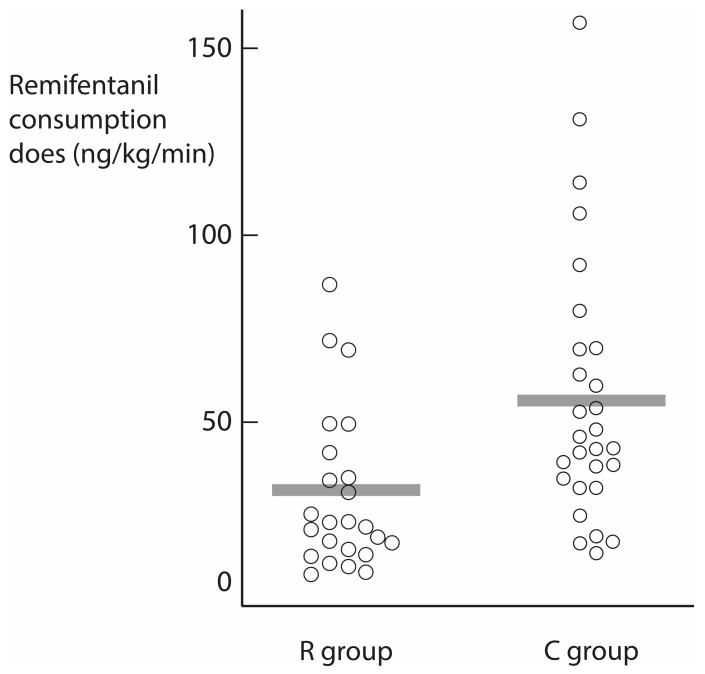
Remifentanil dose distribution between two groups. Gray bars indicate average remifentanil consumption doses of each group.

**Table 1 ijerph-19-16012-t001:** Patient characteristics.

Variable	R Group (24)	C Group (28)	*p*-Value
Age (years)	57 ± 11	51 ± 14	0.095
Sex (male, %)	24 (100%)	27 (96%)	
Height	168 ± 5.6	166 ± 8.1	0.38
Weight	67 ± 15.2	64 ± 11.8	0.376
BMI (kg/m^2^)	23 ± 4.3	23 ± 3.2	0.538
ASA status score			
1	14 (58 %)	23 (82%)	0.73
2	10 (42 %)	5 (18%)	
Initial SPI	51 ± 21.2	49 ± 21.5	0.853
Initial MBP	101 ± 12.4	106 ± 13.0	0.137
HR (bpm)	68 ± 11.1	69 ± 16.0	0.744
Total operation time (min)	55 ± 28.6	51 ± 21.4	0.551
Total anaesthetic time (min)	88 ± 28.6	77 ± 22.6	0.126
Operators			
L	10	13	0.366
C	4	7	
J	11	8	

Abbreviations are as follows: BMI:, body mass index; ASA, American Society of Anaesthesiologist classification; SPI, surgical pleth index; MBP, mean blood pressure; HR, heart rate; R group, regional block group; C group, control group. Values are presented as mean ± SD or number (%).

**Table 2 ijerph-19-16012-t002:** Intraoperative SPI-related data.

	R Group	C Group	*p*-Value
Intraoperative mean SPI	28 ± 7.6	29 ± 9.6	0.763
Intraoperative remifentanil consumption (ng/kg body weight/min)	30 ± 21.2	56 ± 35.5	0.002 *
Remifentanil control frequency	3.0 ± 1.7	4.6 ± 2.2	0.007 *
Intraoperative average mean blood pressure (mmHg)	85 ± 14.7	90 ± 9.3	0.32
Intraoperative average HR (bpm)	64 ± 9.3	58 ± 8.0	0.053
Extubation time (min)	8.6 ± 3.4	9.1 ± 4.3	0.43

Abbreviations are as follows: SPI, surgical pleth index; HR, heart rate; R group, regional block group; C group, control group. Values are presented as mean ± SD. *: *p* < 0.05.

**Table 3 ijerph-19-16012-t003:** Postoperative pain-related data.

	R Group	C Group	*p*-Value
Recovery room discharge time (min)	29.0 (4.02)	31.2 (4.54)	0.071
VAS score			
Resting			
Recovery room	1.0 (0.7)	3.29 (1.15)	0.000 *
4 hours post-surgery	0.96 (0.69)	3.29 (1.24)	0.000 *
8 hours post-surgery	1.09 (0.793)	3.33 (1.27)	0.000 *
Coughing			
Recovery room	1.52 (0.827)	4.61 (1.4)	0.000 *
4 hours post-surgery	1.46 (0.83)	4.32 (1.57)	0.000 *
8 hours post-surgery	1.5 (1.06)	4.41 (1.80)	0.000 *
Additional analgesic, n (%)			
Recovery room	0 (0%)	10 (35%)	0.01 *
4 hours post-surgery	3 (13%)	10 (35%)	0.049 *
8 hours post-surgery	4 (17%)	3 (11%)	0.540
Nausea vomiting scale			
Recovery room	0.13 (0.33)	0.18 (0.39)	0.598
At 8 hours post-surgery	0.07 (0.26)	0.08 (0.26)	0.874

Abbreviations are as follows: VAS, visual analogue scale; R group, regional block group; C group, control group. *: *p* < 0.05, values are presented as mean ± SD or number (%).

## Data Availability

The data presented in this study are available on request from the corresponding author.

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
