# Peer review of "Abdominal Wall Block Decreases Intraoperative Opioid Con-Sumption by Surgical Pleth Index-Guided Remifentanil Administration in Single-Port Laparoscopic Herniorrhaphy: A Prospective Randomized Controlled Trial"

_ijerph, 2022, doi:10.3390/ijerph192316012_

Round 1

Reviewer 1 Report

1.      The language and grammar needs a thorough revision.

Abstract

1.      I would not use the word “preemptive” here

  1. Your conclusion: “SPI can be used to quantify…” is a little bit disturbing to me. You have decided beforehand that your tool for detecting the analgesia provided by the blocs is SPI. Then your conclusion is that your tool can be used. However, you did not use any other tools or validated your tool so you cannot, in my opinion, conclude that your tool can be used. I would delete the last sentence of the abstract.

Introduction

  1. You should more clearly state the aim of the study and your null hypothesis at the end of “Introduction

2.       I do not understand, what you mean in lines 51-52

Methods

1.             Did the patients give their written, informed consent? If no, why?

2.             I am not happy with your nausea and vomiting scale. Usually they are separated as a postoperative patient can vomit without nausea and suffer from severe nausea without vomiting. Please, comment.

3.             There are some things in the Methods section that should be moved to Discussion. For instance, the choice of regional block (lines 161-166).

Results

1.                   Page 7, line 245 and elsewhere. Please, do not use phrase “significant” It is either statistically significant or clinically significant (or both).

2.                   Some of the results are unnecessarily repeated in the text and Tables

3.                   It is not clear to me, what did you do to the data of patients that were excluded during the study. Their data should, in my opinion, be included until they were excluded.

Discussion

1.      Your discussion could be condensed. For instance, you should not report very largely the complications of poor analgesia as you did not study it. Also, the text about SPI could be condensed.

2.      You should rewrite your discussion more clearly. Please, present your main findings and how these findings should be implemented in the clinical practice.

3.      I do not understand your statement line 354

4.      Conclusions: You cannot extrapolate your results to all abdominal blocs. Please, name the blocks you used here.

5.      Also here the sentence in line 358 should be omitted as from the Abstract

6.      Line 360. This is a good sentence

7.   You inserted the blocks while patients were under general anesthesia. You could comment if it is safe (or potential complications) insert this kind of blocks when the patients are not awake. This is controversial at least in many places around the world.

Tables and Figures

1.      You could add a flow chart

2.      You have too many significant numbers in many of your Tables. For instance, your non-invasive blood pressure values are not recorded with decimals. Therefore, they do not become more accurate with calculations (89.62 in Table 2).. Also the Tables are easier to read with less significant numbers.

3.      You should report the units in your Tables. For instance, min, beat/min, cm etc.).

4.      You should have same parametres in Methods and Results section. For instance, I do not find any mention in the Methods about Recovery discharge time measurement nor the criteria for discharge. However, there are results of this in Table 3

References

  1. There are too many references. Please, leave only the most relevant ones.
  2. The references are not expressed according to guidelines of the journal

Author Response

Comments and Suggestions for Authors

  1. The language and grammar needs a thorough revision.

Sorry for making you read my clumsy sentence. There are limitations because the authors are not native speakers.

So, we have already proofread on the language editing services, and we can also show you the certificate of that. However, if it is still insufficient, we will send it to language editing services company contracted with this journal after major revisions are completed.

Abstract

  1. I would not use the word “preemptive” here

Your conclusion: “SPI can be used to quantify…” is a little bit disturbing to me. You have decided beforehand that your tool for detecting the analgesia provided by the blocs is SPI. Then your conclusion is that your tool can be used. However, you did not use any other tools or validated your tool so you cannot, in my opinion, conclude that your tool can be used. I would delete the last sentence of the abstract.

 Thanks for your advice, I deleted the last sentence.

Introduction

You should more clearly state the aim of the study and your null hypothesis at the end of “Introduction”

Thanks for the good point The following sentence was inserted into the introduction part.

We hypothesize that intraoperative opioid dosage would be reduced while maintaining the same nociception level by measuring SPI through preoperative abdominal wall block in SP TEP patients.

  1. I do not understand, what you mean in lines 51-52

I thought there was a leap in the explanation, so I deleted the latter part and organized the contents.

Because the expansion of local anaesthetic is decided by the tension of the fascial layers and muscle in a manner that cannot be replicated in cadaver studies.[10,11] Therefore the anesthesiologist can more effectively discern the amount of pain, the effect of the regional block can be more accurately understood. =>

Because the expansion of local anaesthetic is decided by the tension of the fascial layers and muscle in living tissue, but it cannot be replicated in cadaver studies.[10,11]

Methods

  1. Did the patients give their written, informed consent? If no, why?

We obtained informed consent from all patients and made a description of it. (line 80)

  1. I am not happy with your nausea and vomiting scale. Usually they are separated as a postoperative patient can vomit without nausea and suffer from severe nausea without vomiting. Please, comment.

That's a sharp point. Nausea and vomiting can have different severities. In this paper, since the primary target is the amount of opioid injected during surgery, we decided to use a rather simple scale for severity for N/V. So The n/v 4 point scale described in the following paper was used.

(White, H.; Black, R. J.; Jones, M.; Mar Fan, G. C., Randomized comparison of two anti-emetic strategies in high-risk patients undergoing day-case gynaecological surgery. Br J Anaesth 2007, 98 , (4), 470-6.)

  1. There are some things in the Methods section that should be moved to Discussion. For instance, the choice of regional block (lines 161-166).

The sentence below was appropriate for the Discussion, so it was sent to that part.

(Interestingly, when the local anaesthetic was injected into the medial side, it may interfere with the surgical field of view, so the injection was done in the lateral side as much as possible (Figure 2B). In total, 15 mL of 0.3% ropivacaine was injected on each side.)

Results

  1. Page 7, line 245 and elsewhere. Please, do not use phrase “significant” It is either statistically significant or clinically significant (or both).

We removed the word ‘significant’ and deleted unnecessary content.

  1. Some of the results are unnecessarily repeated in the text and Tables

Repetitive content has been sorted out.

  1. It is not clear to me, what did you do to the data of patients that were excluded during the study. Their data should, in my opinion, be included until they were excluded.

SPI is a new device and I think it needs a lot of development to get a more accurate value. It measures the balance of the sympathetic and parasympathetic nerves. Therefore, there are many factors (HR, position, arrhythmia, type of inhalational anesthetic, peripheral perfusion, inotropic agent) that affect the measurement of SPI. Great care was taken to obtain appropriate results, and if other factors were thought to influence the SPI values, they were excluded from the study.

Discussion

  1. Your discussion could be condensed. For instance, you should not report very largely the complications of poor analgesia as you did not study it. Also, the text about SPI could be condensed.

Unnecessary content has been deleted. It was sent to the introduction to make it easier for people who do not know the contents related to SPI to understand.

  1. You should rewrite your discussion more clearly. Please, present your main findings and how these findings should be implemented in the clinical practice.

I reduced unnecessary content and focused on the content related to the experiment.

  1. I do not understand your statement line 354

Sorry for the sentences you can't understand. I changed it to something like this:

There is an upper limit (ce, 7ng/kg/min) for remifentanil infusion rate, but no lower limit. In other words, even with 0.5 ng/kg/min, when the SPI value was 30 or less, that cases were not excluded from the study. 

  1. Conclusions: You cannot extrapolate your results to all abdominal blocs. Please, name the blocks you used here.

Replaced the following words: ‘Rectus sheath block and quadratus lumborum block’

  1. Also here the sentence in line 358 should be omitted as from the Abstract

Thanks for the good advice, I've removed it

  1. Line 360. This is a good sentence
  2. You inserted the blocks while patients were under general anesthesia. You could comment if it is safe (or potential complications) insert this kind of blocks when the patients are not awake. This is controversial at least in many places around the world.

In the case of a block targeting a direct neural structure, paresthesia can be used to detect nerve damage. Therefore, it is safe to do so without general anesthesia or sedation. However, in the case of interfascial plane block, it does not target nerves, but between planes. General anesthesia is rather advantageous because the patient's breathing is constant and does not respond to needling.

Tables and Figures

  1. You could add a flow chart

We added a Consort flow chart.

  1. You have too many significant numbers in many of your Tables. For instance, your non-invasive blood pressure values are not recorded with decimals. Therefore, they do not become more accurate with calculations (89.62 in Table 2).. Also the Tables are easier to read with less significant numbers.

Units have been changed to make it easier to understand.

  1. You should report the units in your Tables. For instance, min, beat/min, cm etc.).

Unit has been inserted into the table.

  1. You should have same parametres in Methods and Results section. For instance, I do not find any mention in the Methods about Recovery discharge time measurement nor the criteria for discharge. However, there are results of this in Table 3

Thanks for the good point. When designing the study, it was thought that the discharge time would also be different due to the difference in remifentanil infusion dosage amount between the two groups. A score of 10 on the Modified Aldrete score and a score of 9 or higher on the Postanesthesia discharge scoring system (PADS) were planned for discharge. However, the discharge time could not be accurately measured because the pick-up time from the ward was not regular.

Added the following to method:

In the recovery room, when the patient's recovery score reaches 10 on the Modified Aldrete score and 9 or higher on the Postanesthesia discharge scoring system (PADS), the patient is transferred to the ward and the recovery time is recorded.

References

There are too many references. Please, leave only the most relevant ones.

We deleted unnecessary references.

The references are not expressed according to guidelines of the journal

The reference has been revised according to the Journal's guidelines.

Reviewer 2 Report

page 2 p 3-might be better to explain more exactly what SPI is- photoplesmographic analysis of pulse wave and heart beat interval. They do it later but it was confusing not to know then. Perhaps give what values are needed under anesthesia, again done later but seems to go better there. 

Great on same page that is says same person doing all blocks.

P3 good standardization of anesthesia technique

P5 were there block failures?

I think this is great and might think about saying perhaps going forward blocks using Exparel, so longer lasting sparing pain post op would have significant effect on anesthesiologists being able to effect the current opioid crisis.

Author Response

Reviewer 2

page 2 p 3-might be better to explain more exactly what SPI is- photoplesmographic analysis of pulse wave and heart beat interval. They do it later but it was confusing not to know then. Perhaps give what values are needed under anesthesia, again done later but seems to go better there. 

Good point. Added more explanation about SPI at the beginning.

Great on same page that is says same person doing all blocks.

Yes, I'll add something like:

All nerve blocks were performed by a expert with 10 years of experience in regional anesthesia.

P3 good standardization of anesthesia technique

Thank you.

P5 were there block failures?

Thanks for the good point

Since nerve block was performed after general anesthesia, it was possible to check whether the block was successful only after the operation was completed. The rectus sheath block was 100% successful and was written in the 6-page result section.

I think this is great and might think about saying perhaps going forward blocks using Exparel, so longer lasting sparing pain post op would have significant effect on anesthesiologists being able to effect the current opioid crisis.

It's not allowed to be sold in South Korea yet, so I can't use it, but I'll definitely try it later.

Reviewer 3 Report

Presentation of the study is remarkable. I congratulate the reasearhers for that. But you had some serious flaws. Remifentanil end dose during perioperative period in patients with or without preemptive regional block is not a very reliable predictor of outcome for patients. Results with the extubation time, level of pain 4 hours after and discharge time were all similar between the groups. Maybe if used fentanyl or sufentanil it could have different outcomes regarding extubation time, possible side effects of opioids and discharge time. Measuring opioid consumation during preemptive block is of course going to give that result in opioid consumation. In control groups you didnt give any semi-opioid or opioid for that matter in analgesic dose (remifentanil half time is 6 min) so again that control group did well - lower analgesic dosage after 8 hours with R group at discharge time. Serious mistake is giving only NSAID and not analgesic opioid after the operation especially when using ultra short opioids like remifentanil.

Why using 2 blocks tehnique - many reports suggets using TAP or INIH block technique with good results - you are using GA as well so it just for analgesic purposes. From this authors experience try using levobupivacaine rather then ropivacaine as it gives a better sensory block.

Author Response

Presentation of the study is remarkable. I congratulate the reasearhers for that.

But you had some serious flaws.

Remifentanil end dose during perioperative period in patients with or without preemptive regional block is not a very reliable predictor of outcome for patients.

The reviewer's opinion that the dose of remifentanil alone does not have a block effect makes sense.

It turns out that BP, HR, and BIS values, which were expected to represent pain, do not reflect pain well.

Currently, there is no perfect device for measuring nociception in general anesthesia, but it is being developed continuously, and SPI and ANI are the most outstanding devices among them.

When planning the research, I referenced many previous papers related to SPI. The SPI value indicates the degree of nociception, but more precisely, it can be said to be a value related to the sensitivity to feel nociception. If the same dose of remifentanil was maintained and the difference in SPI between the two groups was considered as the target, the method would violate the principle of general anesthesia, which should provide adequate analgesia for nociception. Therefore, as in the example paper, the study was conducted by setting SPI 30, which has the highest sensitivity and specificity, as a target.

Results with the extubation time, level of pain 4 hours after and discharge time were all similar between the groups. Maybe if used fentanyl or sufentanil it could have different outcomes regarding extubation time, possible side effects of opioids and discharge time.

As you already know, the fresh out speed of remifenil and sevo inhalational anesthetics is very fast. Therefore, there was a difference in remifentanil does between the two groups, but it does not seem to affect the extubation time or discharge rate. Also, as the primary target is opioid consumption, strict criteria related to extubation or discharge could not be established.

thanks for the good point I will definitely refer to it for my next study.

Measuring opioid consumption during preemptive block is of course going to give that result in opioid consumation. In control groups you didnt give any semi-opioid or opioid for that matter in analgesic dose (remifentanil half time is 6 min) so again that control group did well - lower analgesic dosage after 8 hours with R group at discharge time. Serious mistake is giving only NSAID and not analgesic opioid after the operation especially when using ultra short opioids like remifentanil.

As a reviewer, you can worry enough about that.

Originally, in single port laparoscopic hernia at our hospital, the use of opioids is restricted because same-day discharge is a rutine protocol. However, if the patient's pain is not resolved with NSAIDs, tramadol is given.

Why using 2 blocks tehnique - many reports suggets using TAP or INIH block technique with good results - you are using GA as well so it just for analgesic purposes. From this authors experience try using levobupivacaine rather then ropivacaine as it gives a better sensory block.

Thank you for the good point. I think INIH block can be a good nerve block method for open inguinal hernia. However, since this operation requires insertion of a laparoscopic instrument from the navel to the inguinal area, a nerve block method that can block a wider area was needed. It is known that QR block can block not only INIH but also T10-L2.

Round 2

Reviewer 3 Report

Note that the protocol in your outpatient hospital dictates using NSAIDs after the procedures or using semi opioids if they alone can not provide sufficient analgesia.

Please do a minor english spell check.

Author Response

Note that the protocol in your outpatient hospital dictates using NSAIDs after the procedures or using semi opioids if they alone can not provide sufficient analgesia.

Thanks for the good point. We added the following:

After surgery, patients follow the routine acute pain service protocol for SP TEP. For those patients with a VAS score of six points or higher in the ward, 100 mg of tramadol was given.

Please do a minor english spell check.

We corrected some spelling.